# Stereodivergent synthesis of chiral amines bearing vicinal stereocenters via hydroamination of trisubstituted alkenes

Haohao Bai[1], Mingchao Li[2], Xiuping Wang[1], Tao Jiang[1], Lintao Zeng[2], Lanlan Zhang [1] & Chao Wang [1] ✉

Chiral aliphatic amines bearing vicinal stereocenters are prevalent motifs in pharmaceuticals and bioactive molecules, yet efficient and stereodivergent access to these structures remains a longstanding challenge. Herein, we report a nickel-catalyzed enantioselective hydroamination of acyclic trisubstituted alkenes that provides a unified platform for the stereodivergent construction of chiral amines bearing β,γ-stereocenters. The reaction exhibits broad substrate scope, accommodating diverse amine electrophiles and tri-substituted alkenes, including those derived from complex bioactive molecules, with high levels of regio-, diastereo-, and enantioselectivity. By modulating the alkene geometry and the configuration of a chiral biimidazoline ligand, all four stereoisomers can be accessed in a predictable manner. The protocol proceeds under mild conditions, tolerates various functional groups, and enables late-stage derivatization, demonstrating its utility for constructing densely functionalized, three-dimensional amine scaffolds. This work provides a valuable platform for asymmetric synthesis and holds strong potential for drug discovery and molecular design.

Chiral aliphatic amines are recognized as one of the most significant classes of organic compounds and serve as privileged scaffolds in pharmaceutical design[1-4]. Considerable efforts have been dedicated to the preparation of enantioenriched aliphatic amines featuring a single C(sp3) stereocenter at various positions (α, β, γ and beyond)[5-14]. Nevertheless, research has shown that incorporating a higher proportion of sp³-hybridized carbons and more stereocenters into drug candidates can significantly boost their clinical success[15,16]. Consequently, synthesizing valuable aliphatic amines with multiple stereogenic centers, which are frequently found in various pharmaceuticals and biologically active natural products (Fig. 1A), is highly desirable. Meanwhile, stereodivergent synthesis is increasingly recognized as essential for advancing asymmetric synthesis and facilitating the discovery of pharmaceuticals and agrochemicals[17,18]. This significance arises from the fact that different enantiomers exhibit distinct phy-

siological properties, which can markedly influence biological function and toxicity[19,20]. Therefore, developing a modular and efficient platform for the stereodivergent synthesis of chiral amines bearing two stereocenters would enhance drug discovery efforts and advance the field of organic synthesis, though it remains exceptionally challenging and elusive.

Enantioselective hydrofunctionalization of alkenes has emerged as a state-of-the-art methodology in asymmetric synthesis and greatly enlarged the library of three-dimensional chiral molecular architechtures[21-24]. We envisioned that the stereoselective hydroamination[25-28] of acyclic trisubstituted alkenes would serve as an ideal and versatile approach for the stereodivergent synthesis of chiral amines bearing vicinal stereocenters (Fig. 1B)[29-45]. Given that olefin migratory insertion occurs in a syn fashion, this strategy can enable the access to all four stereoisomers by adjusting the geometry of the

[1]Tianjin Key Laboratory of Structure and Performance for Functional Molecules; College of Chemistry, Tianjin Normal University, Tianjin, PR China. [2]School of Light Industry and Food Engineering, Guangxi University, Nanning, PR China. ✉e-mail: chwang@tjnu.edu.cn

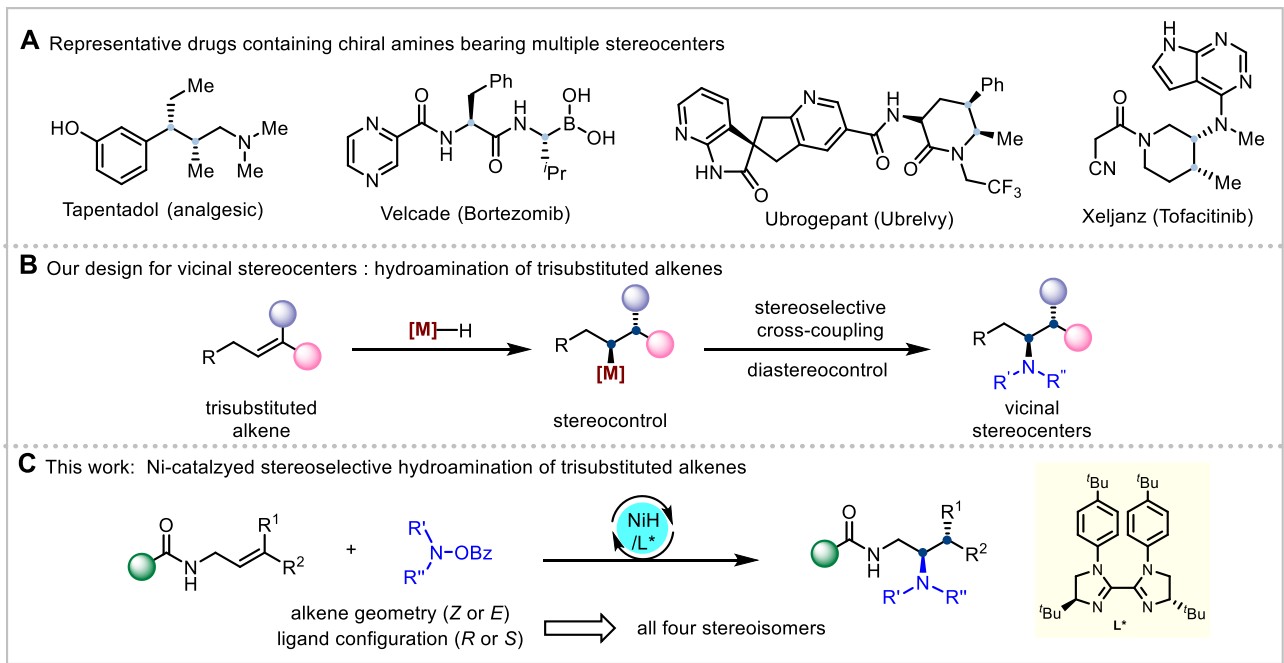

**Fig. 1 | Diastereo- and enantioselective hydroamination of trisubstituted alkenes. A** Representative drugs containing chiral amines bearing multiple stereocenters. **B** Our design for vicinal stereocenters: hydroamination of trisubstituted alkenes. **C** This work: Ni-catalzyed stereoselective hydroamination of trisubstituted alkenes.

alkene and the configuration of the chiral ligand, representing a distinct chiral induction model from stereodivergent dual catalysis[46–56]. However, realizing this blueprint entails several formidable challenges. First, steric congestion in trisubstituted olefins diminishes their binding affinity toward transition metal catalysts, thereby reducing the efficiency of migratory insertion. Moreover, regioselectivity control is often difficult due to the absence of stabilizing adjacent polar groups, which can lead to undesired migratory hydrofunctionalization products[57–61]. Finally, achieving precise control over both the absolute configurations of each stereocenter and their relative stereo-chemistry remains particularly challenging.

In light of these challenges, we envisioned that combining a strategically positioned monodentate directing group with a judiciously selected chiral ligand within a single nickel catalytic system could offer a viable solution[62–65]. In this study, we develop a practical and broadly applicable nickel-catalyzed protocol for the enantioselective hydroamination of acyclic trisubstituted alkenes, enabling stereodivergent access to enantioenriched three-dimensional amines bearing vicinal β,γ-stereocenters (Fig. 1C). To our knowledge, examples of constructing contiguous stereocenters from acyclic trisubstituted alkenes via NiH-catalyzed asymmetric hydrofunctionalization are limited, and this study establishes a general strategy to address this challenge. Notably, by modulating both the geometry of the alkene (E or Z) and the absolute configuration of the chiral biimidazoline ligand (R or S), all four stereoisomers can be selectively accessed in a predictable manner. This approach not only addresses the longstanding challenges of regio-, diastereo-, and enantioselective control in highly congested trisubstituted alkenes[66–71], but also provides a versatile strategy for the programmable construction of complex chiral amines, which would accelerate the development of densely functionalized amine scaffolds for applications in medicinal chemistry.

## Results and discussion
### Evaluation of reaction conditions
Our investigation toward this nickel-catalyzed enantioselective hydroamination reaction commenced by selecting (E)-γ-methyl-γ-phenylethyl-substituted N-benzoyl allyl amide (**1a**), and morpholino benzoate (**2a**) as model components to evaluate and optimize the

reaction parameters (Table 1). A series of chiral ligands were initially screened using NiBr₂•DME as the nickel source, LiOH as the base, and ᵗBuOH as the solvent at 30 °C for 12 h. Common classes of privileged chiral ligands, including, bioxazoline (**L1–L3**), pyridine-oxazoline (**L4**), quinoline-imidazoline (**L5**), bis(oxazoline) (**L6**), phosphine-oxazoline (**L7**), diamine (**L8**), and various diphosphine ligands, all proved ineffective in promoting the desired asymmetric transformation. To our delight, biimidazoline ligands (**L9–L11**) exhibited promising performance, delivering high yield, excellent diastereoselectivity, and outstanding enantioselectivity. In particular, **L12** enabled the formation of hydroamination product **3a** as a single diastereomer in 90% isolated yield, with >99% ee and >20:1 dr (entry 1). Alternative nickel sources such as Ni(OTf)₂ or NiBr₂ led to significantly diminished yields (entries 2–3). Among the silanes evaluated, Me(EtO)₂SiH was identified as the optimal hydrogen source (entries 4–5). Further screening of bases revealed that LiOH provided the best yield and enantioselectivity (entries 6–8). The addition of KI was found to enhance both yield and ee, probably by promoting the regeneration of the Ni–H species (entries 9–10)[72]. Notably, the choice of solvent proved critical, as ᵗBuOH was essential for achieving high reactivity and enantioselectivity (entries 11–12). When the reaction temperature was lowered to 10 °C, a reduced yield was obtained while maintaining comparable enantioselectivity (entry 13), likely due to partial solidification of the solvent at this temperature.

### Substrate scope
With the optimized conditions in hand, we next explored the scope of amine electrophiles in the asymmetric hydroamination of trisubstituted alkenes (Fig. 2). In general, both cyclic and acyclic O-benzoylhydroxylamine (N–O) reagents proved to be competent substrates, affording chiral 1,2-diamine products with excellent levels of enantio- and diastereoselectivity. Various tetrahydropyridine and piperazine derivatives bearing functional groups such as esters, acetals, carbamates, and sulfones were well tolerated (**3b–3h**). Notably, the synthesis of **3i** with a pyrimidine motif underscores the method's tolerance of additional nitrogen donors without sacrificing reactivity or selectivity. Our protocol was further expanded to acyclic secondary amine electrophiles (**3j–3y**). The steric bulk of the alkyl substituents on

**Table. 1 | Variation of reaction parameters[a]**

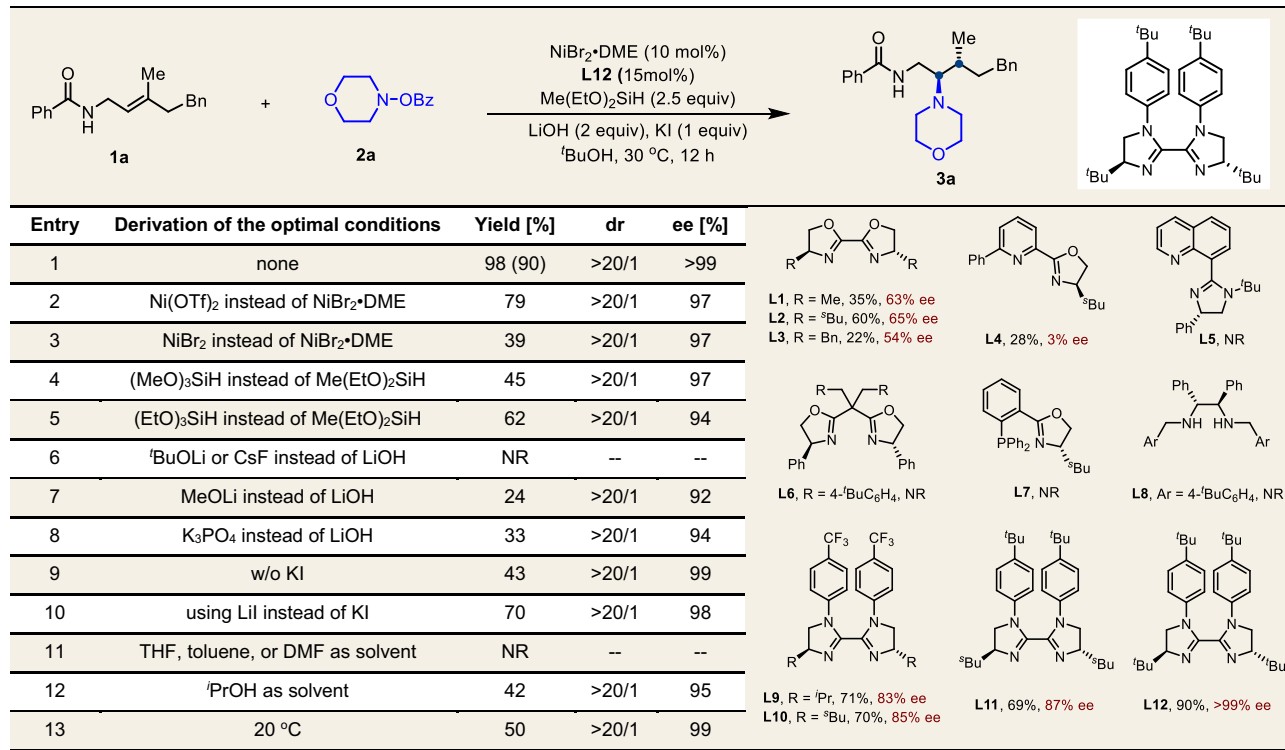

| Entry | Derivation of the optimal conditions | Yield [%] | dr | ee [%] |
|---|---|---|---|---|
| 1 | none | 98 (90) | >20/1 | >99 |
| 2 | Ni(OTf)$_2$ instead of NiBr$_2$•DME | 79 | >20/1 | 97 |
| 3 | NiBr$_2$ instead of NiBr$_2$•DME | 39 | >20/1 | 97 |
| 4 | (MeO)$_3$SiH instead of Me(EtO)$_2$SiH | 45 | >20/1 | 97 |
| 5 | (EtO)$_3$SiH instead of Me(EtO)$_2$SiH | 62 | >20/1 | 94 |
| 6 | $^t$BuOLi or CsF instead of LiOH | NR | -- | -- |
| 7 | MeOLi instead of LiOH | 24 | >20/1 | 92 |
| 8 | K$_3$PO$_4$ instead of LiOH | 33 | >20/1 | 94 |
| 9 | w/o KI | 43 | >20/1 | 99 |
| 10 | using LiI instead of KI | 70 | >20/1 | 98 |
| 11 | THF, toluene, or DMF as solvent | NR | -- | -- |
| 12 | $^i$PrOH as solvent | 42 | >20/1 | 95 |
| 13 | 20 °C | 50 | >20/1 | 99 |

Conditions[a]: **1a** (0.2 mmol) and **2a** (0.3 mmol) were used in $^t$BuOH (1 mL); Yields were determined by the isolated products; Isolated yields within the parentheses. THF: Tetrahydrofuran, DMF: *N,N*-Dimethylformamide.

the amine moiety had little effect on either reactivity or enantioselectivity (**3j–3m**). Moreover, alkyl chlorides remained intact under the reaction conditions (**3p** and **3q**), offering opportunities for downstream diversification via cross-coupling[73,74]. A variety of dibenzyl-substituted hydroxylamine electrophiles bearing versatile functional groups such as fluoride, chloride, thioether, trifluoromethoxy, and boronic ester were well tolerated (**3u–3y**), delivering products with high reactivity and enantioselectivity. This result highlights the potential of the method for drug-oriented synthesis and late-stage derivatization.

Subsequently, we evaluated the substrate scope of trisubstituted alkenes (Fig. 3). We first examined various amide groups on γ,γ-disubstituted allyl amides and found that both electron-rich and electron-deficient aromatic as well as aliphatic amides were well tolerated under the standard conditions, affording products bearing vicinal stereocenters with high efficiency (**4a–4i**). Functionalized aryl and heteroaryl groups, including cyano-, nitro-substituted phenyl and thiophene, were also compatible. The absolute configuration was determined by X-ray crystallography of **4b** (CCDC: 2403087) as (*R, R*). The nature of the R$^2$ substituent on the trisubstituted alkene had negligible impact on reaction efficiency or selectivity. Substrates bearing benzyl groups (**4j–4n**), short-chain alkyls such as propyl (**4o**), and long-chain alkyls (**4p–4r**) all underwent smooth hydroamination with excellent regio-, enantio-, and diastereoselectivity. Alkenes derived from natural products such as phytol (**4r**) and citronellol (**4 s**) also furnished the desired products with high selectivity. Notably, alkyl chains containing silyl ether (**4j** and **4u**) or acetate (**4v**) groups were also compatible, enabling further downstream derivatization. Furthermore, switching the R$^1$ substituent from methyl to ethyl or isopropyl required ligand adjustment and still delivered the desired product (**4w, 4x**) with high enantioselectivity, highlighting the catalyst's ability to differentiate between similar alkyl groups. In addition, exocyclic alkenes also participated in the hydroamination to deliver the desired product (**4y** and

**4z**) in good yield with high enantioselectivity, while the unsymmetric counterpart (**4aa**) reacted with diminished yield but retained high e.e. The *N*-methylated allylamide failed to react, which may indicate that the N–H moiety contributes to facilitating olefin insertion, potentially through noncovalent interactions as previously suggested[71].

To assess the potential of this hydroamination strategy for stereodivergent synthesis, we examined whether all four stereoisomers could be selectively accessed by systematically varying the olefin geometry and ligand configuration (Fig. 4). Remarkably, the transformation of **3v** proceeded smoothly under each set of conditions, delivering all four stereoisomers in high yields with high levels of enantio- and diastereocontrol. These results demonstrate that our protocol enables precise and programmable access to any desired stereoisomer within a single catalytic platform. Such flexibility is particularly valuable for applications in medicinal chemistry, where stereochemical diversity can play a crucial role in modulating biological activity.

To further illustrate potential synthetic utility of our protocol in medicinal chemistry, the late-stage derivatization of natural products and drug-like molecules was conducted (Fig. 5A). The alkenes derived from Febuxostat and Isoxepac with **2a** effectively underwent asymmetric hydroamination to afford products (**5a** and **5b**) in high yields with excellent ee and diastereoselectivity. Moreover, amino electrophiles derivatized from Atomoxetine, Duloxetine, Maprotiline, and Nortriptyline, reacted smoothly to deliver corresponding products (**5c–5f**) with moderate to good yields. We further demonstrated the practicability and synthetic value of this enantioselective protocol through a gram-scale reaction and product derivatizations (Fig. 5B). Delightfully, the catalytic system was successfully scaled up to a 5.0 mmol scale, leading to the formation of **3a** in 78% yield with excellent enantio- and diastereoselectivity. The amide group could be deprotected by NaOH at 120 °C, delivering the corresponding free primary amine (**6a**) without loss of enantioselectivity. Reduction of the amide group with lithium aluminum hydride provided the secondary

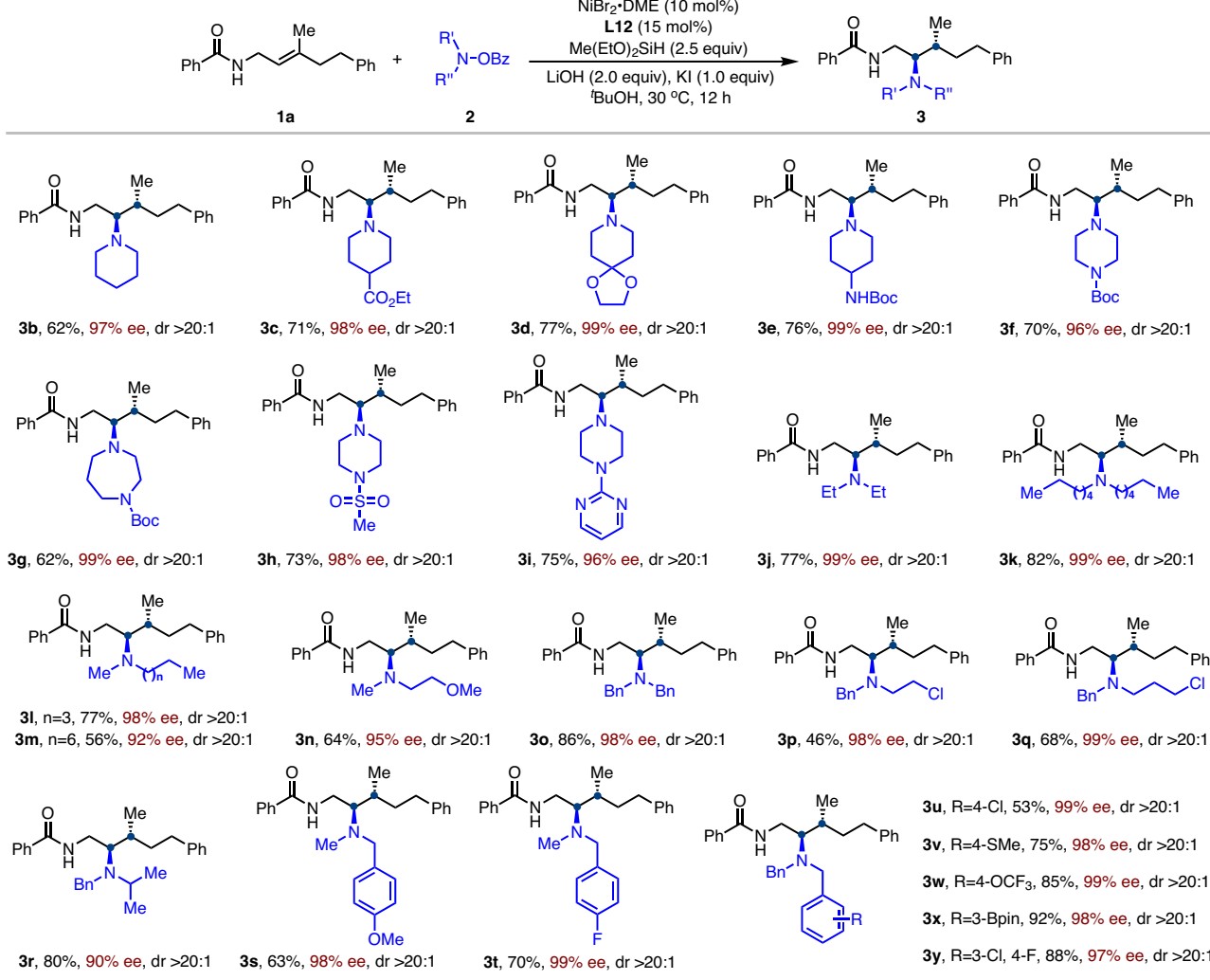

**Fig. 2 | Scope of amine electrophiles[a].** [a]Conditions: **1a** (0.2 mmol) and **2** (0.3 mmol) were used in [t]BuOH (1 mL); Yields were determined by the isolated products, dr was determined by NMR analysis of the crude products.

amine (**6b**) with 98% e.e. Moreover, benzoyl amide could be smoothly converted to *N*-Boc-amide (**6c**) and thioamide (**6d**), respectively. *N*-(2-ethylamino)amide scaffolds are structural motifs frequently found in biologically active molecules, including the marketed drugs Moclobemide and Procainamide (Fig. 5C). To further highlight the relevance and synthetic potential of our protocol, we prepared two stereochemically enriched analogs of these drugs (**7a** and **7b**), both containing two additional stereocenters.

## Mechanistic studies

To gain insight into the mechanism of this asymmetric hydroamination, we conducted a series of preliminary mechanistic studies (Fig. 6). When terminal alkene **8a** and 1,2-disubstituted alkene **8b** were subjected to the standard conditions (Fig. 6A), the corresponding β-selective products **9a** and **9b** were obtained with slightly diminished enantio- and regioselectivity. These results suggest that the γ-substituent plays a favorable steric role in controlling both regio- and enantioselectivity. Deuterium-labeling experiments using a deuterated silane with terminal or trisubstituted alkenes revealed exclusive deuterium incorporation at the γ-position (Fig. 6B), indicating that the hydride originates from the silane and that the hydrometallation step is stereospecific and irreversible. Moreover, the use of deuterium-labeled alkene **1a-d2** gave the product **3a-d2** without any H/D scrambling, supporting that the alkyl–Ni intermediate couples directly with

the N–O electrophile without undergoing a isomerization process. The addition of butylated hydroxytoluene (BHT) as a radical scavenger had minimal impact on the reaction efficiency or selectivity, suggesting that a radical mechanism is unlikely to be operative (Fig. 6C). To gain some information on the active nickel catalyst, a positive linear correlation was observed between the ee values of the chiral ligand and the corresponding hydroamination product (Fig. 6D), implying that the catalytically active nickel species likely bears a single biimidazoline ligand. Furthermore, a well-defined Ni(II) complex (**L12·NiBr₂**) was synthesized and characterized by X-ray crystallography (Fig. 6E), which confirmed the coordination geometry. Notably, this complex served as a competent catalyst, delivering **3a** in comparable yield and enantioselectivity. Finally, HRMS (ESI) analysis of the reaction mixture revealed the presence of Ni(II)−H and Ni(I)−X species ligated with the chiral ligand (Fig. 6F), offering direct evidence for their existence and providing valuable mechanistic insight into the NiH catalytic cycle. These findings represent a meaningful contribution to the broader understanding of nickel hydride catalysis.

Based on the above experimental results and related literature precedents[32,75], a plausible catalytic cycle is proposed in Fig. 6G. The Ni(II) precatalyst (**A**), in the presence of a hydrosilane, a chiral ligand, and a base, could be converted into a mixture of Ni(II)−H species (**B**) and a Ni(I)−X species (**C**)[76,77]. The Ni(II)−H species undergoes migratory insertion with the trisubstituted alkene under the assistance of the

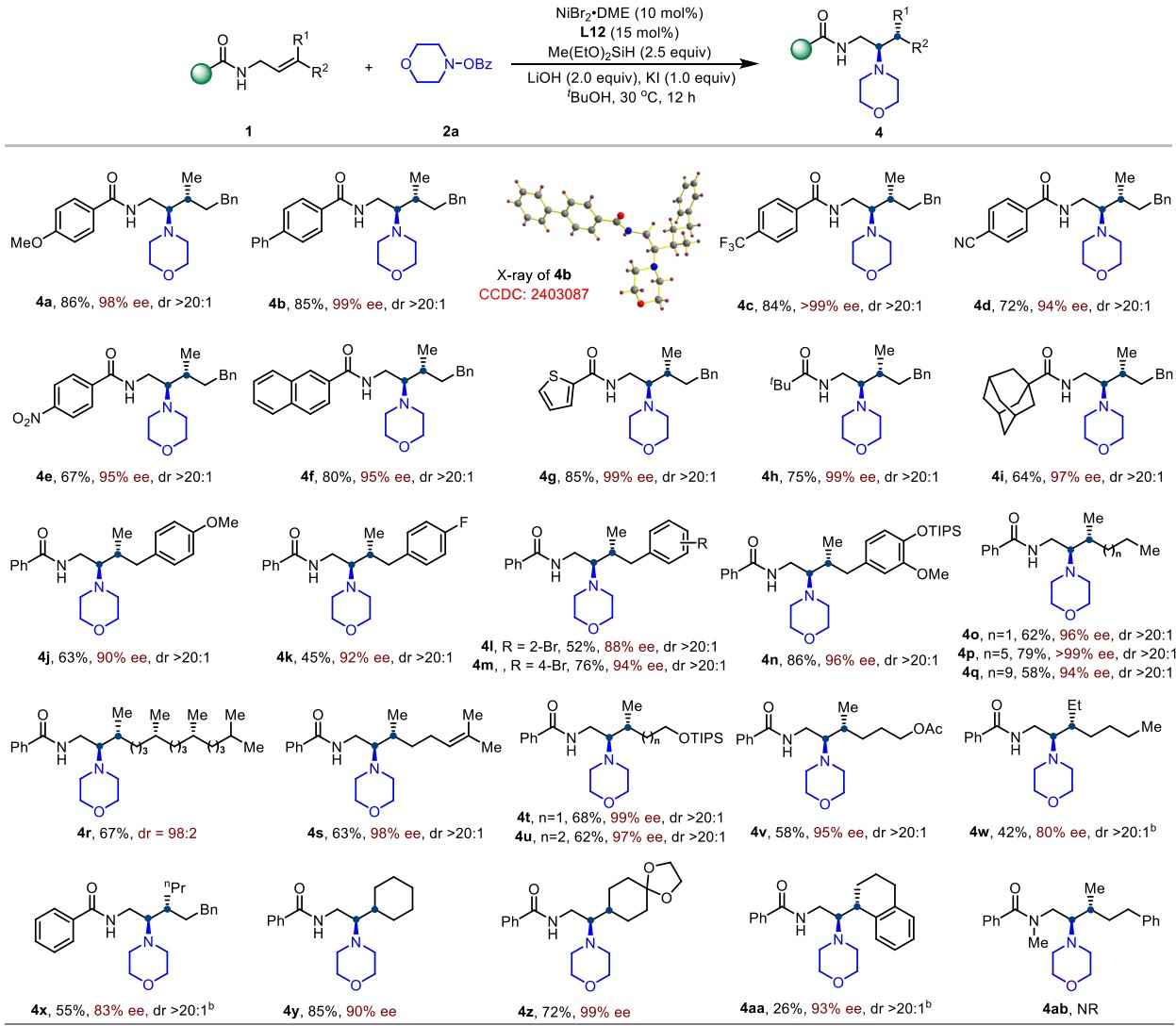

**Fig. 3 | Scope of trisubstituted alkenes[a].** [a]Conditions: **1** (0.2 mmol) and **2a** (0.3 mmol) were used in [t]BuOH (1 mL); Yields were determined by the isolated products, dr was determined by NMR analysis of the crude products. [b]Conditions: **L2** instead of **L12**.

amide group, to form a Ni(II)−alkyl intermediate (**D**). This intermediate then undergoes transmetalation with the Ni(I)−X species[75,78], generating a Ni(I)−alkyl intermediate (**E**). Electrophilic interception of **E** by the N−O electrophile, followed by reductive elimination, furnishes the hydroamination product **3** along with regeneration of the Ni(I)-X species (X = OBz). The Ni(II)−H catalyst is then regenerated via the reaction of the resulting Ni(II)−X₂ species with osilane, thus completing the catalytic cycle. Moreover, we cannot rule out the possibility of transmetalation between Ni(II)−H and Ni(I)−H species. Although Ni(I)−H intermediates were not detected by high-resolution mass spectrometry (HRMS), presumably due to their inherent instability, they could plausibly arise from the reaction of Ni(I)−X with hydrosilane[79,80].

In summary, we have developed a nickel-catalyzed enantioselective hydroamination of acyclic trisubstituted alkenes that enables the stereodivergent synthesis of chiral aliphatic amines bearing vicinal stereocenters. This strategy exhibits high levels of regio-, diastereo-, and enantioselectivity across a broad array of amine electrophiles and alkene substrates, including those derived from complex bioactive molecules. By tuning the alkene geometry and the configuration of the chiral ligand, all four stereoisomers can be selectively accessed within a single catalytic platform. Mechanistic studies revealed the presence of Ni(II) and

Ni(I) species, and suggest that a transmetalation step between them plays a pivotal role in the catalytic cycle. We anticipate that the developed protocol will greatly enrich the toolbox of asymmetric catalysis for the synthesis of dual stereocenters, and accelerate the discovery and optimization of drug candidates and complex functional molecules.

## Methods
### Procedure for the enantioselective hydroamination of trisubstituted alkenes
In an argon-filled glovebox, NiBr₂•DME (6.2 mg, 0.02 mmol, 10.0 mol %), Ligand (15.4 mg, 0.03 mmol, 15 mol%), relevant alkene substrate (0.2 mmol, 1.0 equiv), LiOH (9.6 mg, 0.4 mmol, 2.0 equiv), KI (33.2 mg 0.2 mmol, 1.0 equiv), [t]BuOH (1.0 mL) were added to a 4 mL reaction flask. Then relevant hydroxylamine ester (0.3 mmol, 1.5 equiv), Me(EtO)₂SiH (80 µL, 0.5 mmol, 2.5 equiv) were added to the mixture. The reaction mixture was stirred at 30 °C for 12 h. After the reaction time, remove solvent under vacuum. Remaining liquid was allowed to silica gel column chromatography. The crude product was purified by column chromatography on silica gel with a mixture of ethyl acetate and petroleum ether as eluent. More experimental details and characterization are available in Supplementary Information.

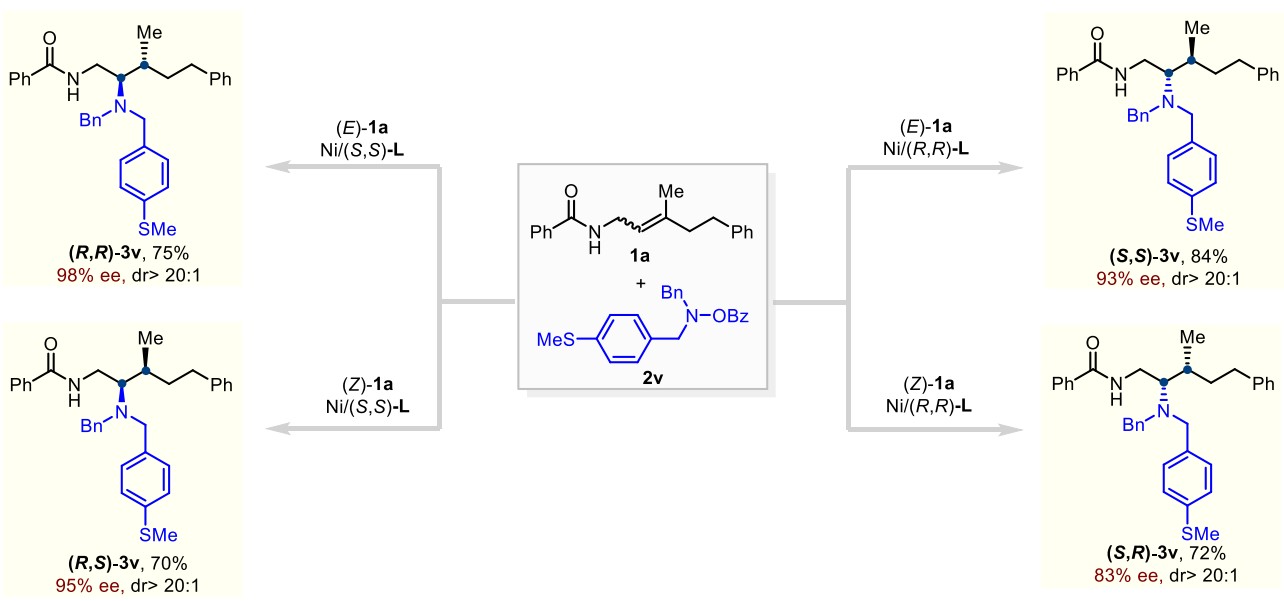

**Fig. 4 | Stereodivergent synthesis.** Access to all stereoisomers by systematically varying the olefin geometry and ligand configuration. Isolated yields are reported and diastereomeric ratios (dr) were determined by NMR analysis of the crude reaction mixture.

**A)** Late-stage decoration of drug-like molecules

**5a**, 73%, 97% ee, dr >20:1
from Febuxostat derivative

**5b**, 45%, 98% ee, dr >20:1
from Isoxepac derivative

**5c**, 46%, dr >99:1
from Atomoxetine derivative

**5d**, 55%, dr >99:1
from Duloxetine derivative

**5e**, 65%, >99% ee, dr >20:1
from Maprotiline derivative

**5f**, 61%, 97% ee, dr >20:1
from Nortriptyline derivative

**B)** Gram-scale reaction and further derivatization

**6a**, 91%, 98% ee

**6b**, 85%, 98% ee

NaOH
EtOH
120 °C

(Boc)₂O
DMAP
MeCN, rt

**3a**, 5 mmol, 1.42 g
78%, 98% ee

LiAlH₄
THF, 70 °C

Lawesson's
Reagent
PhMe, 85 °C

**6c**, 78%, 98% ee

**6d**, 93%, 98% ee

**C)** Construction of drug analogues featuring *N*-(2-ethylamino)amide motifs

**1aa**     **2a**

NiBr₂•DME (10 mol%)
**L12** (15 mol%)
Me(EtO)₂SiH (2.5 equiv)
LiOH (2.0 equiv), KI (1.0 equiv)
ᵗBuOH, 30 °C, 12 h

**7a**, 75%, 99% ee, dr >20:1

Moclobemide
(Antidepressant)

**1ab**     **2j**

NiBr₂•DME (10 mol%)
**L12** (15 mol%)
Me(EtO)₂SiH (2.5 equiv)
LiOH (2.0 equiv), KI (1.0 equiv)
ᵗBuOH, 30 °C, 12 h

**7b**, 45%, 92% ee, dr >20:1

Acecainide
(Antiarrhythmic)

**Fig. 5 | Product derivatizations. A** Late-stage decoration of drug-like molecules. **B** Gram-scale reaction and further derivatization. **C** Construction of drug analogues featuring N-(2-ethylamino)amide motifs.

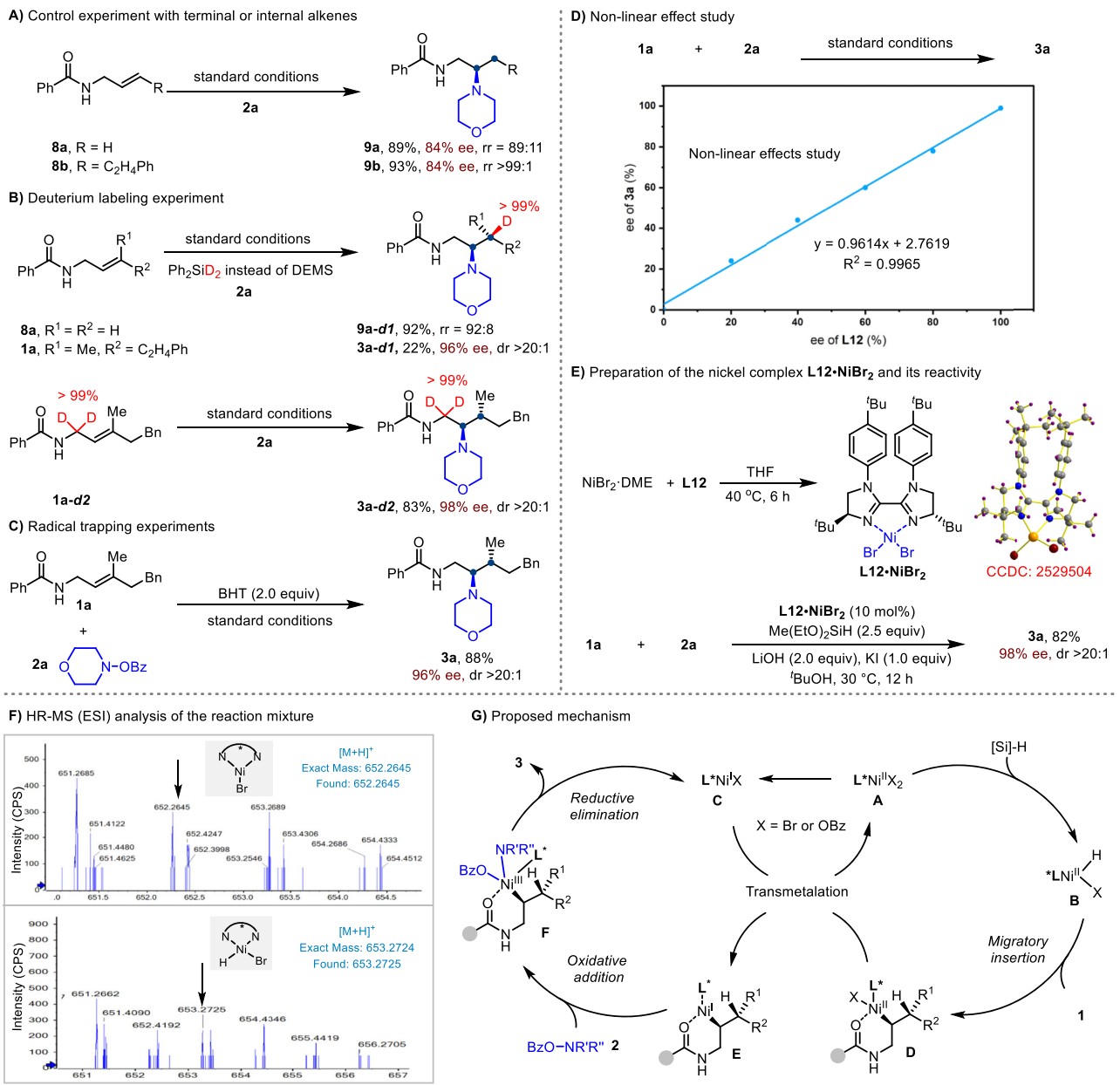

**Fig. 6 | Mechanistic studies. A** Control experiment with terminal or internal alkenes. **B** Deuterium-labeled experiment. **C** Radical trapping experiments. **D** Non-linear effect study. **E** Preparation of the nickel complex L12•NiBr$_2$ and its reactivity. **F** HR-MS (ESI) analysis of the reaction mixture. **G** Proposed mechanism.

## Data availability

All data needed to evaluate the conclusions in the paper are present in the paper and/or the Supplementary Information. Additional data related to this paper are available from the corresponding authors upon request. The supplementary crystallographic data for this paper are available free of charge from the Cambridge Crystallographic Data Centre(CCDC) under accession numbers CCDC 2403087 (compound **4b**), CCDC 2529504 (compound **L12•NiBr$_2$**). Copies of the data can be obtained free of charge via https://www.ccdc.cam.ac.uk/structures/.

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

## Acknowledgements

This work was supported by the National Natural Science Foundation of China (22301216 C.W. and 22571231 C.W.), the Natural Science Foun-dation of Tianjin (23JCYBJC00760 to C.W.), Tianjin Normal University Research Innovation Project for Postgraduate Students (2024KYCX054Z to H.B.), and funds provided by Tianjin Normal University.

## Author contributions

H.B. carried out most of the experiments, M.L., X.W. and T.J. helped to separate and purify some target products, L.Z. (LinTao Zeng), L.Z. (Lan-lan Zhang) and C.W. wrote the manuscripts, C.W. designed the project and directed the work.

## Competing interests

The authors declare no competing interests.
