## [Transparent Peer Review file · Nature Communications]

Stereodivergent Synthesis of Chiral Amines bearing Vicinal Stereocenters via Hydroamination of Trisubstituted Alkenes

Corresponding Author: Professor Chao Wang

Version 0:

Reviewer comments:

Reviewer #1

(Remarks to the Author)

In this manuscript, Wang and co-workers report a nickel-catalyzed enantioselective hydroamination of trisubstituted alkenes, enabling the construction of chiral amines bearing β,γ -stereocenters. The reaction displays a broad substrate scope across various amine electrophiles and trisubstituted alkenes, delivering products with high levels of regio-, diastereo-, and enantioselectivity. The synthetic value of this methodology is further highlighted by stereodivergent access to all four possible stereoisomers, achieved through modulation of alkene geometry and the chirality of the biimidazole ligand. The transformation proceeds under mild conditions, exhibits good functional group tolerance, and is applicable to late-stage functionalization.

Overall, this work represents a synthetically valuable advance, particularly in light of the longstanding challenges associated with the enantioselective construction of amine scaffolds bearing vicinal stereocenters. Accordingly, this reviewer recommends publication in Nature Communications, provided that the following minor concerns are adequately addressed.

- The Flack parameter of compound **4b** is reported as -1.0 (Supplementary Information, Supplementary Figure 6, S91). While this reviewer is not an expert in X-ray crystallography, it is recognized that such a value may indicate that the assigned absolute configuration should be inverted relative to that shown. The authors are encouraged to re-examine this issue. X-ray crystallographic analysis of substrates bearing pre-existing chirality (e.g., compound **4r**) might provide more reliable confirmation of the absolute configuration.
- Only β,γ -unsaturated substrates have been explored. Is it possible to extend the carbon chain between the amide and alkene functionalities, and if so, how does this affect reactivity and selectivity?
- Is the amide functionality essential for controlling regio- and enantioselectivity? While directing groups are often crucial for regio-control, selectivity in the present system may instead be governed largely by steric effects, with nickel preferentially inserting into the monosubstituted alkene site. Although secondary amides were shown to be ineffective, investigation of other functional groups would be informative.
- A notable limitation from a synthetic perspective is that a methyl group appears to be the only effective substituent on the disubstituted carbon. If a small substituent is required for effective enantiodiscrimination, could other small substituents—such as heteroatoms (e.g., fluorine)—be tolerated?
- The data provided in the Supplementary Information are generally of good quality; however, several ^{13}C NMR spectra exhibit very weak signals, making it difficult to confidently assess the purity of the isolated compounds. Higher-quality NMR spectra should be provided.

Reviewer #2

(Remarks to the Author)

Wang and co-authors report an enantioselective nickel catalyzed hydroamination of acyclic trisubstituted alkenes to generate chiral amines bearing β,γ -stereocenters. The reaction demonstrates excellent regioselectivity, enantioselectivity, diastereoselectivity and potential synthetic utility, delivering a diverse array of products in moderate to good yields. A series of mechanistic studies are conducted including the control experiments, deuterium-labeling experiments, radical trapping experiment and NLE study, et. al, and the authors propose that the hydroamination reaction proceeds through synergistic Ni(I)/Ni(II) cycles with a key transmetalation step between them. Overall, the paper is well written and portrayed nicely.

However, with respect to the question of novelty and impact, there is a strong past precedent of asymmetric hydroamination of alkenes via Cu-H or Ni-H catalysis as shown below:

<https://pubs.acs.org/doi/10.1021/ja4092819>; <https://pubs.acs.org/doi/10.1021/ja509786v>;

<https://onlinelibrary.wiley.com/doi/full/10.1002/anie.201410326>;

<https://onlinelibrary.wiley.com/doi/full/10.1002/anie.201803026>;

<https://onlinelibrary.wiley.com/doi/full/10.1002/anie.202109881>;

<https://pubs.acs.org/doi/10.1021/acscatal.2c02892>;

<https://pubs.acs.org/doi/full/10.1021/acscatal.3c01845>;

<https://pubs.rsc.org/en/Content/ArticleLanding/2025/QO/D4QO02275K>;

https://pubs.acs.org/doi/pdf/10.1021/jacs.4c03854?ref=article_openPDF;

<https://pubs.acs.org/doi/full/10.1021/acs.orglett.3c02442>.

There is also an analogous enantioselective example of this reaction via Ni-H catalysis (Ref. 26 cited in the manuscript) which is almost certainly mechanistically related.

As it stands, this manuscript is not suitable for publication in the Nature Communication.

Reviewer #3

(Remarks to the Author)

The manuscript entitled "Stereodivergent Synthesis of Chiral Amines bearing Vicinal Stereocenters via Hydroamination of Trisubstituted Alkenes" is authors' continuing research on NiH-catalyzed net hydrofunctionalization of alkenes with N-O electrophiles. They systematically investigated the reaction parameters, including catalyst, ligand, base, and solvent, to identify the effective system for the promotion of titled reaction. The observed yield, regioselectivity, diastereoselectivity, and enantioselectivity all are extremely high, thus delivering the enantioenriched chiral aliphatic amines with multiple stereocenters. In particular, the catalyst- and substrate geometry-controlled stereodivergent approach to all possible stereoisomers deserves significant attention.

However, this reviewer has a critical concern on its novelty. In the literature, there are many successful examples of NiH-catalyzed hydrofunctionalizations of alkenes with the assistance of native directing groups such as amine, amide, and alcohol. Of course, this reviewer agrees with the authors' claim that the trisubstituted alkene still remains a challenging substrate, but the present optimal catalyst is the almost same as the authors' previous work (ACS Catal. 2023, 13, 10041). The authors state that the presently developed biimidazoline ligand shows uniquely high performance, but this is not new (e.g., JACS 2021, 143, 15873) and seems to be just minor modification of previous bioxazoline system. Such routine work cannot meet the criteria of novelty and urgency for publication in Nat. Commun.

On the basis of the above considerations, this reviewer recommends transfer to more specialized synthetic journals such as ACIE and ChemistryEurope after the following revision.

- 1) Seminal work and recent comprehensive review-type articles on metal-hydride-mediated net hydroamination of alkenes should be mentioned and cited: ACIE 2013, 52, 10830; JACS 2013, 135, 15746; ACIE 2016, 55, 48; JACS 2022, 144, 648.
- 2) Many biologically active amines contain the cyclic skeleton rather than the acyclic one. Thus, the authors should also try the cyclic alkene substrate.
- 3) The stereodivergent synthesis of all four possible stereoisomers within a single catalyst system deserves significant attention, but the precedent based on CuH should also be fairly mentioned and cited: Nature 2016, 532, 353.
- 4) In Table 1, the authors observed significant impacts of base and solvent on the reaction efficiency, but their effects are not discussed at all in the part of mechanistic studies.
- 5) In Table 3, various monodentate amide groups are compatible. This reviewer wonders if the bidentate amide such as picoline amide is also be tolerated.
- 6) In the synthesis of 4r (Table 3), this reviewer wonders if the match-mismatch phenomenon is observed with ent-L12.
- 7) Page 5, line 104; "tertiary amines" rather than "secondary amines"?
- 8) In Figure 4G and related discussion parts, the authors use "transmetalation", but is it correct? "electron transfer" is better?
- 9) minor errors:
page 8, line 156; "L" of Lithium aluminum hydride should be lowercase.
page 8, line 162; insert a space between additional and stereocenters.

Reviewer #4

(Remarks to the Author)

Version 1:

Reviewer comments:

Reviewer #1

(Remarks to the Author)

While NiH-catalyzed hydrofunctionalization of alkenes has been broadly studied, the concomitant installation of two stereocenters via trisubstituted alkenes remains underexplored, as it necessitates a distinct catalytic approach. In this revised manuscript, the authors have successfully addressed the comments from the previous round of reviews, further

strengthening a manuscript that presents a novel route to 1,2-stereocenters from trisubstituted alkenes. As such, this reviewer recommends this manuscript for publication in Nature Communications, provided that the following minor corrections are further made.

- The authors should correct '(4x and 4z)' to '(4y and 4z)' on page 6, line 131.
- The authors noted in their response letter that several unsuccessful substrates (4ac, 4ad–4aj) were added to the SI. However, this reviewer was unable to locate these examples in the revised version. The authors should verify these results and include them in the final SI.

Reviewer #3

(Remarks to the Author)

I have reviewed the original submission of this manuscript. First, I do appreciate the authors' efforts to address my concerns. However, also as commented by the reviewer 2, I do not think that this work meets the criteria of novelty and urgency for publication in Nat. Commun. because there are many precedents of NiH- and related CuH-catalyzed stereoselective net hydroamination of alkenes.

Submission to more synthetic journals is recommended.

First of all, we acknowledge the four reviewers for their valuable comments they made to the title manuscript. The point-by-point responses to the reviewer's comments are listed *in italic text in blue*.

Reviewer 1

In this manuscript, Wang and co-workers report a nickel-catalyzed enantioselective hydroamination of trisubstituted alkenes, enabling the construction of chiral amines bearing β,γ -stereocenters. The reaction displays a broad substrate scope across various amine electrophiles and trisubstituted alkenes, delivering products with high levels of regio-, diastereo-, and enantioselectivity. The synthetic value of this methodology is further highlighted by stereodivergent access to all four possible stereoisomers, achieved through modulation of alkene geometry and the chirality of the biimidazoline ligand. The transformation proceeds under mild conditions, exhibits good functional group tolerance, and is applicable to late-stage functionalization.

Overall, this work represents a synthetically valuable advance, particularly in light of the longstanding challenges associated with the enantioselective construction of amine scaffolds bearing vicinal stereocenters. Accordingly, this reviewer recommends publication in Nature Communications, provided that the following minor concerns are adequately addressed.

Response: We sincerely thank the reviewer for the positive and encouraging assessment of our work. We are delighted that the reviewer recognizes the significance of our nickel-catalyzed enantioselective hydroamination of trisubstituted alkenes and its synthetic value in accessing amines bearing β,γ -stereocenters with high levels of stereocontrol. We are also grateful for the reviewer's recommendation for publication in Nature Communications.

1. The Flack parameter of compound 4b is reported as -1.0 (Supplementary Information, Supplementary Figure 6, S91). While this reviewer is not an expert in X-ray crystallography, it is recognized that such a value may indicate that the assigned absolute configuration should be inverted relative to that shown. The authors are encouraged to re-examine this issue. X-ray crystallographic analysis of substrates bearing pre-existing chirality (e.g., compound 4r) might provide more reliable confirmation of the absolute configuration.

Response: We thank the reviewer for raising this important point regarding the Flack parameter of compound 4b. We have re-refined the crystallographic data, and the Flack parameter of 4b is now -0.1 , which supports the originally assigned absolute configuration. The revised crystallographic data and updated parameters have been included in the Supplementary Information. Regarding the suggestion to perform X-ray crystallographic analysis on substrates bearing pre-existing chirality (e.g., compound 4r), we note that these compounds were obtained as an oil and did not yield suitable single crystals for X-ray diffraction despite multiple attempts. Importantly, under the present reaction conditions, the

stereocenter located on the substituent is not involved in bond formation or cleavage and is therefore generally expected to remain unchanged.

Supplementary Table 6. Crystal data and structure refinement for **4b**.

Identification code	a250225a
Empirical formula	C ₂₉ H ₃₄ N ₂ O ₂
Formula weight	442.58
Temperature/K	293(2)
Crystal system	monoclinic
Space group	P2 ₁ (4)
a/Å	8.7469(5)
b/Å	5.5489(3)
c/Å	25.1685(14)
α/°	90
β/°	93.461(5)
γ/°	90
Volume/Å ³	1219.35(11)
Z	2
ρ _{calc} /cm ³	1.205
μ/mm ⁻¹	0.588
F(000)	476
Crystal size/mm ³	0.12×0.13×0.25
Radiation	Cu K _α (λ=1.54184 Å)
2θ range for data collection/	7.04 to 134.14 (0.84 Å)
Index ranges	-10 ≤ h ≤ 10, -6 ≤ k ≤ 6, -30 ≤ l ≤ 25
Reflections collected	8194
Independent reflections	4240 [R _{int} = 0.0442, R _{sigma} = 0.0433]
Data/restraints/parameters	4240/1/294
Goodness-of-fit on F ²	1.030
Final R indexes [I ≥ 2σ (I)]	R ₁ = 0.0528, wR ₂ = 0.1416
Final R indexes [all data]	R ₁ = 0.0692, wR ₂ = 0.1564

Largest diff. peak/hole / e Å⁻³

0.15/-0.14

Flack parameter

-0.1(3)

2. Only β,γ -unsaturated substrates have been explored. Is it possible to extend the carbon chain between the amide and alkene functionalities, and if so, how does this affect reactivity and selectivity?

Response: We thank the reviewer for this insightful question. We did explore such substrates; however, no desired hydroamination products were observed. We attribute this lack of reactivity to the increased distance between the amide directing group and the alkene, which likely weakens the effective coordination and directional control required for productive Ni-H insertion and subsequent functionalization. These results suggest that a suitably positioned directing group is crucial for achieving both reactivity and stereocontrol in the present catalytic system. We have clarified these unreactive substrates in the revised Supplementary Information.

3. Is the amide functionality essential for controlling regio- and enantioselectivity? While directing groups are often crucial for regio-control, selectivity in the present system may instead be governed largely by steric effects, with nickel preferentially inserting into the monosubstituted alkene site. Although secondary amides were shown to be ineffective, investigation of other functional groups would be informative.

Response: We thank the reviewer for this insightful question. Control experiments indicate that the amide functionality is essential for achieving both reactivity and selectivity in the present system. When alternative functional groups, including carboxylic acid derivatives and ethers, were examined in place of the amide, no desired hydroamination products were observed. These results suggest that regio- and enantioselectivity are not governed solely by steric effects. Instead, the amide directing group may participate in noncovalent interactions with the nickel catalyst, which assist alkene migratory insertion and stereocontrol, as recently demonstrated in related systems (Ref. 61). We have included these unreactive examples in the revised Supplementary Information.

4. A notable limitation from a synthetic perspective is that a methyl group appears to be the only effective substituent on the disubstituted carbon. If a small substituent is required for effective enantiodiscrimination, could other small substituents—such as heteroatoms (e.g., fluorine)—be tolerated?

Response: We thank the reviewer for this insightful comment. We have tested substrates with ethyl and propyl substituents at the disubstituted carbon, and good results were obtained, although ligand adjustment was required to maintain high enantioselectivity. The relevant results and discussion have been included in the revised manuscript. Specifically, as described in the manuscript: switching the R^1 substituent from methyl to ethyl or isopropyl required ligand adjustment and still delivered the desired products (**4w**, **4x**) with high enantioselectivity, highlighting the catalyst's ability to differentiate between closely related alkyl groups.

5. The data provided in the Supplementary Information are generally of good quality; however, several ^{13}C NMR spectra exhibit very weak signals, making it difficult to confidently assess the purity of the isolated compounds. Higher-quality NMR spectra should be provided.

Response: We thank the reviewer for pointing this out. In response, we have re-recorded the relevant ^{13}C NMR spectra with longer acquisition times to enhance signal quality, and the updated data have been included in the revised Supplementary Information.

¹³C NMR (101 MHz, CDCl₃) spectra of **3b**

¹³C NMR (101 MHz, CDCl₃) spectra of **3j**

¹³C NMR (101 MHz, CDCl₃) spectra of 3u

¹³C NMR (101 MHz, CDCl₃) spectra of **3k**

Reviewer 2

Wang and co-authors report an enantioselective nickel catalyzed hydroamination of acyclic trisubstituted alkenes to generate chiral amines bearing β,γ -stereocenters. The reaction demonstrates excellent regioselectivity, enantioselectivity, diastereoselectivity and potential synthetic utility, delivering a diverse array of products in moderate to good yields. A series of mechanistic studies are conducted including the control experiments, deuterium-labeling experiments, radical trapping experiment and NLE study, et. al, and the authors propose that the hydroamination reaction proceeds through synergistic Ni(I)/Ni(II) cycles with a key transmetalation step between them. Overall, the paper is well written and portrayed nicely.

However, with respect to the question of novelty and impact, there is a strong past precedent of asymmetric hydroamination of alkenes via Cu-H or Ni-H catalysis as shown below:

<https://pubs.acs.org/doi/10.1021/ja4092819>;

<https://pubs.acs.org/doi/10.1021/ja509786v>;

<https://onlinelibrary.wiley.com/doi/full/10.1002/anie.201410326>;

<https://onlinelibrary.wiley.com/doi/full/10.1002/anie.201803026>;
<https://onlinelibrary.wiley.com/doi/full/10.1002/anie.202109881>;
<https://pubs.acs.org/doi/10.1021/acscatal.2c02892>;
<https://pubs.acs.org/doi/full/10.1021/acscatal.3c01845>;
<https://pubs.rsc.org/en/Content/ArticleLanding/2025/QO/D4QO02275K>;
https://pubs.acs.org/doi/pdf/10.1021/jacs.4c03854?ref=article_openPDF;
<https://pubs.acs.org/doi/full/10.1021/acs.orglett.3c02442>.

There is also an analogous enantioselective example of this reaction via Ni-H catalysis (Ref. 26 cited in the manuscript) which is almost certainly mechanistically related.

As it stands, this manuscript is not suitable for publication in the Nature Communication.

Response: *We thank the reviewer for the careful reading and for their positive comments on the clarity, writing, and mechanistic studies presented in our manuscript. We also appreciate the thoughtful discussion regarding prior work on asymmetric hydroamination. We note that the cited examples primarily focus on the formation of a single stereocenter from mono- or disubstituted alkenes. In contrast, as discussed in the Introduction of our manuscript, our work addresses the more challenging problem of constructing vicinal β,γ -stereocenters from acyclic trisubstituted alkenes, enabling stereodivergent access to all four stereoisomers of enantioenriched three-dimensional amines. This is achieved by combining a strategically positioned monodentate directing group with a carefully chosen chiral biimidazoline ligand, and by modulating the geometry of the alkene (*E* or *Z*). To our knowledge, examples of NiH-catalyzed asymmetric hydrofunctionalization that construct contiguous stereocenters from acyclic trisubstituted alkenes are exceedingly limited. We have therefore clarified in the manuscript that our study establishes a general and broadly applicable strategy for addressing this long-standing synthetic challenge.*

Reviewer 3

The manuscript entitled “Stereodivergent Synthesis of Chiral Amines bearing Vicinal Stereocenters via Hydroamination of Trisubstituted Alkenes” is authors' continuing research on NiH-catalyzed net hydrofunctionalization of alkenes with N-O electrophiles. They systematically investigated the reaction parameters, including catalyst, ligand, base, and solvent, to identify the effective system for the promotion of titled reaction. The observed yield, regioselectivity, diastereoselectivity, and enantioselectivity all are extremely high, thus delivering the enantioenriched chiral aliphatic amines with multiple stereocenters. In

particular, the catalyst- and substrate geometry-controlled stereodivergent approach to all possible stereoisomers deserves significant attention.

However, this reviewer has a critical concern on its novelty. In the literature, there are many successful examples of NiH-catalyzed hydrofunctionalizations of alkenes with the assistance of native directing groups such as amine, amide, and alcohol. Of course, this reviewer agrees with the authors' claim that the trisubstituted alkene still remains a challenging substrate, but the present optimal catalyst is the almost same as the authors' previous work (ACS Catal. 2023, 13, 10041). The authors state that the presently developed biimidazoline ligand shows uniquely high performance, but this is not new (e.g., JACS 2021, 143, 15873) and seems to be just minor modification of previous bioxazoline system. Such routine work cannot meet the criteria of novelty and urgency for publication in Nat. Commun.

On the basis of the above considerations, this reviewer recommends transfer to more specialized synthetic journals such as ACIE and ChemistryEurope after the following revision.

Response: *We thank the reviewer for the careful reading of our manuscript and for the positive comments regarding the high yield, regio-, diastereo-, and enantioselectivity, as well as the stereodivergent control achieved in this work.*

Regarding the novelty concern, while we agree that NiH-catalyzed hydrofunctionalizations of alkenes with native directing groups have been reported, these prior examples primarily focus on monosubstituted or disubstituted alkenes and typically generate a single stereocenter. In contrast, our study addresses the challenging class of acyclic trisubstituted alkenes and enables the construction of vicinal β,γ -stereocenters, which has remained largely unexplored. The biimidazoline ligand, though structurally related to previously reported systems, exhibits uniquely high performance for these highly congested trisubstituted alkenes, enabling stereodivergent access to all four stereoisomers by modulating both the alkene geometry (E/Z) and the ligand configuration. To our knowledge, this combination of substrate type, contiguous stereocenter construction, and predictable stereodivergence has not been demonstrated in previous NiH-catalyzed hydrofunctionalization studies.

1. Seminal work and recent comprehensive review-type articles on metal-hydride-mediated net hydroamination of alkenes should be mentioned and cited: ACIE 2013, 52, 10830; JACS 2013, 135, 15746; ACIE 2016, 55, 48; JACS 2022, 144, 648.

Response: *We thank the reviewer for highlighting these important review articles, which have now been cited in the revised manuscript as Ref. 25–28.*

2. Many biologically active amines contain the cyclic skeleton rather than the acyclic one. Thus, the authors should also try the cyclic alkene substrate.

Response: *We thank the reviewer for this helpful suggestion. Cyclic alkene substrates have been examined in this study, and they underwent the*

hydroamination smoothly to afford the corresponding cyclic amine products with good yields and high levels of regio- and enantioselectivity (see products 4x, 4z, and 4aa). These results demonstrate that the present catalytic system is also applicable to cyclic alkene substrates.

3. The stereodivergent synthesis of all four possible stereoisomers within a single catalyst system deserves significant attention, but the precedent based on CuH should also be fairly mentioned and cited: Nature 2016, 532, 353.

Response: We thank the reviewer for this important suggestion. The seminal CuH-catalyzed stereodivergent approach reported in Nature (2016, 532, 353) has now been cited in the revised manuscript as Ref. 66.

4. In Table 1, the authors observed significant impacts of base and solvent on the reaction efficiency, but their effects are not discussed at all in the part of mechanistic studies.

Response: We thank the reviewer for this insightful comment. We agree that the base and solvent have a pronounced influence on the reaction efficiency, as reflected in Table 1. At the present stage, we attribute these effects mainly to their roles in facilitating the generation and modulating the reactivity of the Ni–H species, as well as stabilizing key organonickel intermediates along the catalytic cycle. In particular, the base is likely involved in the formation of reactive Ni(I)/Ni(II)–H species under the reaction conditions. As direct experimental evidence is currently unavailable, we refrain from overinterpreting these observations. A discussion focusing on the role of the base has been added to the revised manuscript to clarify this point.

5. In Table 3, various monodentate amide groups are compatible. This reviewer wonders if the bidentate amide such as picoline amide is also be tolerated.

Response: We thank the reviewer for this insightful question. We examined bidentate directing groups, including picolinamide and 8-aminoquinoline amide; however, no desired hydroamination products were observed. We anticipate that the strong bidentate coordination occupies the nickel coordination sphere and prevents effective binding of the chiral ligand, thereby reducing the reactivity of the catalyst. These results further underscore the importance of a monodentate amide directing group in enabling both reactivity and enantioselectivity in the present system. Relevant results have been included in the revised Supplementary Information.

6. In the synthesis of 4r (Table 3), this reviewer wonders if the match-mismatch phenomenon is observed with ent-L12.

Response: We thank the reviewer for this thoughtful question. When the (R,R)-L12 was employed in the synthesis of 4r, comparable yield and diastereoselectivity were obtained relative to those observed with (S,S)-L12, indicating that no obvious match–mismatch effect is present in this case. These results suggest that the pre-existing stereocenter in the substrate does not strongly interfere with the catalyst-controlled stereochemical outcome. Relevant data have been included in the revised Supplementary Information.

7. Page 5, line 104; "tertiary amines" rather than "secondary amines"?

Response: We thank the reviewer for pointing this out. The phrase “secondary amines” has been revised to “secondary amine electrophiles” in the revised manuscript.

8. In Figure 4G and related discussion parts, the authors use "transmetalation", but is it correct? "electron transfer" is better?

Response: We thank the reviewer for this insightful comment. The use of the term “transmetalation” is supported by well-established precedents in nickel catalysis involving Ni(I)/Ni(II) species. In many NiH - mediated hydrofunctionalization reactions, a transmetalation-type process is commonly invoked to describe the net transfer of an organic fragment between Ni(I) and Ni(II) intermediates, even if electron transfer may also occur. For example, ligand-controlled NiH migratory hydrofunctionalization has been described using this terminology (Chem 2021, 7, 3171–3188), and mechanistic studies of reductively induced aryl transmetalation in nickel-catalyzed cross-coupling explicitly invoke Ni(I)-to-Ni(II) transmetalation pathways (Romero-Arenas et al., J. Am. Chem. Soc. 2025, 147, 21697–21707). These examples illustrate that “transmetalation” is widely used to denote the net exchange of organic groups between nickel species of different oxidation states, and does not preclude concomitant electron transfer. Relevant references have now been added in the revised manuscript (Refs. 75 and 80).

9. minor errors:

page 8, line 156; "L" of Lithium aluminum hydride should be lowercase.

page 8, line 162; insert a space between additional and stereocenters.

Response: We thank the reviewer for pointing these out. We have corrected these errors.

Reviewer 4

Response: We thank Reviewer 4 for their participation in evaluating our manuscript.

First of all, we acknowledge the two reviewers for their valuable comments they made to the title manuscript. The point-by-point responses to the reviewer's comments are listed *in italic text in blue*.

Reviewer 1

While NiH-catalyzed hydrofunctionalization of alkenes has been broadly studied, the concomitant installation of two stereocenters via trisubstituted alkenes remains underexplored, as it necessitates a distinct catalytic approach. In this revised manuscript, the authors have successfully addressed the comments from the previous round of reviews, further strengthening a manuscript that presents a novel route to 1,2-stereocenters from trisubstituted alkenes. As such, this reviewer recommends this manuscript for publication in Nature Communications, provided that the following minor corrections are further made.

Response: We sincerely thank the reviewer for the positive evaluation and constructive comments. We are glad that our revisions have strengthened the manuscript. All suggested minor corrections have been carefully implemented in the revised version.

1. The authors should correct '(4x and 4z)' to '(4y and 4z)' on page 6, line 131.

Response: We thank the reviewer for pointing this out. The text has been corrected accordingly.

2. The authors noted in their response letter that several unsuccessful substrates (4ac, 4ad – 4aj) were added to the SI. However, this reviewer was unable to locate these examples in the revised version. The authors should verify these results and include them in the final SI.

Response: The examples of the unsuccessful substrates have been added to Section 5: “Unsuccessful substrates” of the Supplementary Information

Reviewer 2

I have reviewed the original submission of this manuscript. First, I do appreciate the authors' efforts to address my concerns. However, also as commented by the reviewer 2, I do not think that this work meets the criteria of novelty and urgency for publication in Nat. Commun. because there are many precedents of NiH- and related CuH-catalyzed stereoselective net hydroamination of alkenes.

Submission to more synthetic journals is recommended.

Response: We thank Reviewer 2 again for their review.